# A New Conformal Penetrating Heating Strategy for Atherosclerotic Plaque

**DOI:** 10.3390/bioengineering10020162

**Published:** 2023-01-26

**Authors:** Hongying Wang, Shiqing Zhao, Jincheng Zou, Aili Zhang

**Affiliations:** School of Biomedical Engineering, 400 Med-X Research Institute, Shanghai Jiao Tong University, 1954 Huashan Road, Shanghai 200030, China

**Keywords:** atherosclerosis, radiofrequency volumetric heating, convection cooling, bidirectional ablation, precise thermal treatment

## Abstract

(1) Background: A combination of radiofrequency (RF) volumetric heating and convection cooling has been proposed to realize plaque ablation while protecting the endothelial layer. However, the depth of the plaque and the thickness of the endothelial layer vary in different atherosclerotic lesions. Current techniques cannot be used to achieve penetrating heating for atherosclerosis with two targets (the specified protection depth and the ablation depth). (2) Methods: A tissue-mimicking phantom heating experiment simulating atherosclerotic plaque ablation was conducted to investigate the effects of the control parameters, the target temperature (*T_target_*), the cooling water temperature (*T_f_),* and the cooling water velocity (*V_f_*). To further quantitatively analyze and evaluate the ablation depth and the protection depth of the control parameters, a three-dimensional model was established. In addition, a conformal penetrating heating strategy was proposed based on the numerical results. (3) Results: It was found that *T_target_* and *T_f_* were factors that regulated the ablation results, and the temperatures of the plaques varied linearly with *T_target_* or *T_f_*. The simulation results showed that the ablation depth increased with the *T_target_* while the protection depth decreased correspondently. This relationship reversed with the *T_f_*. When the two parameters *T_target_* and *T_f_
*were controlled together, the ablation depth was 0.47 mm–1.43 mm and the protection depth was 0 mm–0.26 mm within 2 min of heating. (4) Conclusions: With the proposed control algorithm, the requirements of both the ablation depth and the endothelium protection depth can be met for most plaques through the simultaneous control of *T_target_* and *T_f_*.

## 1. Introduction

Restenosis remains a challenge in treating atherosclerosis with percutaneous transluminal angioplasty (PTA) [1,2,3,4]. The main reason for restenosis is the endothelial injuries caused by the mechanical force of balloon or stent expansion and the alteration of hemodynamics due to stent struts as well as the consequent proliferation of smooth muscle cells (SMCs) [1,5]. Thermal ablation can safely and efficiently remove unwanted tissue [6,7] and has also been used in the treatment of atherosclerosis [8,9,10]. Studies have demonstrated that radiofrequency (RF) balloon angioplasty can effectively ablate atherosclerotic plaque, open occluded vessels, and fuse dissected vascular media [11]. However, restenosis persists since current thermal techniques inevitably overheat the thin endothelium [10,12].

New thermal balloon angioplasty has been proposed to realize the selective heating of atherosclerotic plaque [13]. A radiofrequency microelectrode array is embedded on the surface of a balloon for penetrating volume heating of the plaque tissue, while the circulating cooling agent inside the balloon provides surface convection cooling to protect the endothelial layer [14]. The feasibility of the selective heating strategy was confirmed, but how to ablate plaques of different thicknesses remains unknown. To achieve a successful treatment for plaques of different thicknesses, it is important to precisely control the two modalities at the same time. When the RF volumetric heating power is excessive, overheating occurs and damages the endothelium and the surrounding nerves, which causes restenosis and other secondary injuries [2,15], while overprotection of the endothelium and surrounding tissues results in insufficient plaque ablation.

Successful heating strategies to achieve the desired ablation depth while protecting the thin endothelial layer are quite different from the conformal radiofrequency ablation of tumors. Tumor ablation needs to precisely control the far end of the ablation and sufficiently cover the distal margin of the tumor (several centimeters away from the probe) while sparing the surrounding tissues or important organs. This was achieved through the optimization of the heating power or electrode configuration design [16,17]. For the ablation of atherosclerotic plaques, two parameters, the penetrating ablation depth and the protection depth, need to be satisfied at the same time.

In addition, the thickness of the endothelial layer and the depths of the atherosclerotic plaques vary greatly in different atherosclerotic lesions due to different levels of stenosis and media hyperplasia [18,19,20]. As shown in Figure 1, a typical plaque includes a necrotic core, proliferative SMCs and an overlying intima [3,21]. The necrotic core and the proliferative SMCs are the unwanted tissue and need to be ablated, while the intima, containing a thin endothelial layer, needs to be protected. The intima–media thickness (IMT), which describes the plaque thickness, is approximately 0.55–0.95 mm in peripheral arteries [22] and 0.81–1.01 mm in the femoral artery [18], and the intimal thickness ranges from 0.05 mm to 0.12 mm [23]. The required ablation depth (*A*) and the protection depth (*P*)*,* as defined in Figure 1, are different in every lesion. A conformal ablation strategy for atherosclerotic plaques with various thicknesses that protects the endothelium, which also has different dimensions, is necessary. The target temperature for proliferated SMCs is over 50 °C [24,25], while the temperature in the whole layer of the intima should be under 50 °C, which means that different protection depth requirements and different thermal gradients are needed in the intima.

Therefore, in this study, the influence of the control parameters on heat generation and heat transfer in the newly proposed radiofrequency balloon angioplasty was investigated both experimentally and numerically. Based on the relationship between the ablation results and the control parameters, a control algorithm was proposed to determine the feasibility of bidirectional control for realizing the conformal heating of plaques.

## 2. Materials and Methods

### 2.1. Tissue-Mimicking Phantom Heating Experiment

#### 2.1.1. Experimental Setup

A tissue-mimicking phantom, which consisted of 5.5% gelatin, 0.46% NaCl powder, and deionized water [26], was used to mimic arterial tissue. As shown in Figure 2, a polyimide catheter (inner diameter: 2.5 mm, outer diameter: 2.94 mm) simulating an angioplasty balloon was inserted into the tissue-mimicking phantom, which was placed in a 37 °C thermostat water bath (DK-8A, Shanghai Jinghong Experimental Equipment Co., Ltd., Shanghai, China). To achieve the bidirectional depth control of the penetrating thermal treatment, the bipolar model was chosen. Therefore, each time, only one pair of electrodes (Figure 3a) was used to deliver the 460 kHz radiofrequency energy to the phantom, though there were 3 × 4 electrodes (width and spacing: 1 mm, length: 4 mm) attached to the outer surface of the catheter. The water inside the low-temperature thermostat bath was pumped into the catheter through a soft silicone tube by an adjustable-speed pump (DC40D-2480A, Zhongke Electromechanical Co., Ltd., Shenzhen, China). The host computer controlled the homemade radiofrequency source [27], which was calibrated with an oscilloscope (Agilent InfiniiVision, MSO-X 2022A, Keysight Technologies, Inc., Santa Rosa, CA, USA), and the adjustable-speed pump via a data acquisition device (DAQ, USB3102A, ART Technology, Beijing, China).

Three calibrated temperature sensors (TC1, TC2, and TC3) (K-Type thermocouple, 44AWG, Sandvik, Sweden) were inserted into the tissue-mimicking phantom parallel to the catheter to monitor the phantom temperature in the depth direction (see Figure 3a). They were positioned vertically above the middle of the two working electrodes using the holder with holes. The phantom with thermocouples was photographed from the top and side views before and after 2 min of heating. The accurate positions of the thermocouples were determined using the image processing software Image J. The distances between TC1, TC2, and TC3 and the outer surface of the catheter were 0.85 ± 0.09 mm, 2.46 ± 0.17 mm, and 4.08 ± 0.13 mm. Therefore, TC1, TC2, and TC3 could represent the locations of plaque, adventitia, and surrounding tissue, respectively [28]. Another thermocouple, TC0 (K-Type thermocouple, 44AWG, Sandvik, Sweden), was located on the inner surface of the tissue phantom and represented the inner surface of the blood vessel. TC0 was positioned in the middle of the two working electrodes to measure the temperature at the control point *T_target_* (Figure 3b), and TC0 was isolated with polyimide film (thickness: 0.05 mm) to avoid electromagnetic interference. Similarly, to reduce electromagnetic interference and minimize interference in temperature measurement, TC1, TC2, and TC3 were insulated with thin polyimide tubes (inner diameter: 0.2 mm, thickness: 0.02 mm), and their hot junction tips were covered with thermally conductive silicone grease (STARS-922, Balance Stars, China), which has good electrical insulation and thermal conductivity (>0.671 W/(m∙K)).

#### 2.1.2. Heating Conditions

To protect the endothelial layer from overheating, the temperature-controlled mode is chosen, and the adjustable parameters included the target temperature (*T_target_*), the cooling water temperature (*T_f_*), and the cooling water velocity (*V_f_*). Specifically, *T_target_* is a parameter that adjusts the RF heating power, while *T_f_
*and *V_f_* are parameters that adjust the convective cooling power. To determine the key parameters for ablation control, the influence of these parameters was investigated. Three groups of experiments were designed to study the influence of the parameters on the temperature profile inside the phantom:

*Group one*: *T_target_* was set at 35 °C, 38 °C, 40 °C, 43 °C, 45 °C, and 48 °C. *T_f_* was 15 °C, and *V_f_* was 2.85 m/s.

*Group two*: *T_f_* was adjusted from 10 °C to 30 °C in 5 °C intervals with *T_target_* set at 38 °C and *V_f_* set at 2.85 m/s.

*Group three*: *V_f_* ranged from 1.51 m/s to 4.57 m/s in steps of 0.76 m/s, while *T_target_* was 38 °C and *T_f_* was 20 °C. The corresponding volume flow (Q) ranged from 7.4 ml/s to 22.43 ml/s, and the convective heat transfer coefficient ranged from 7221 W/(m^2^∙K) to 21,745 W/(m^2^∙K), with an average interval of 3631 W/(m^2^∙K). The convective heat transfer coefficient (*h*) was calculated according to the Gnielinski formula [29] (Appendix A, Equations (A1)–(A5)), which was also used in the numerical simulation.

Considering the short duration of several minutes in the real procedure of conventional balloon angioplasty [25,30], the whole experiment included two stages: a 45-s expansion of the balloon using cooling water and 2 min of convection cooling combined with RF heating. Temperatures (T1, T2, and T3) were measured every 500 ms using thermocouples (TC1, TC2, and TC3) with Keysight DAQ970A. Temperature T0 served as the control point. Each experiment was repeated three times.

### 2.2. Numerical Simulation

#### 2.2.1. Model Geometry

To further analyze the relationship between the ablation dimensions and the treatment parameters, an applicable three-dimensional model was established to obtain the temperature and SAR (specific absorption rate) distribution. Different from the radiofrequency catheter ablation in cardiac arrhythmias, such as in [31], the bioheat trans
fer model coupled with an RF electric field and cooling water circulating inside a catheter was studied. At the same time, the temperature dependences of the tissue’s electrical conductivity and thermal conductivity were considered in our model, which was different from previous studies [32,33].

The three-dimensional (3D) geometry was established in COMSOL Multiphysics (COMSOL, Inc., Burlington, MA, USA). A 50 × 50 × 24 mm^3^ block and a hollow cylinder with an inner diameter of 2.5 mm and an outer diameter of 2.94 mm in the middle were used to simulate the tissue and the catheter (Figure 4a). As the fibrotic plaque was mainly composed of proliferating SMCs, which have similar properties to the arterial wall, the fibrotic plaque and the arterial wall were regarded as homogenous [34]. The paired copper electrodes (thickness: 25 μm, width: 1 mm, length: 4 mm) with a spacing of 1 mm were distributed circumferentially on the outer layer of the polyimide catheter. To ensure the precision of the calculation, a finer mesh (minimum size: 0.075 mm, maximum size: 1 mm) was used in the fan-shaped zone close to the working electrodes and the tissue (Figure 4b). In total, there were 380,825 tetrahedral mesh elements, 50,862 triangular mesh elements, and 1650 edge mesh elements in our model. More detailed mesh information is provided in Appendix A (Table A1 and Figure A4).

#### 2.2.2. Mathematical Modeling

For low frequencies, as used in RFA (460 kHz), Maxwell’s equation can be simplified as the quasi-static approach, Equation (1): (1)$$\nabla \cdot \left[\sigma \nabla V\right]=0\;$$
where σ is the electrical conductivity (S/m) and *V* is the electrical potential (V) in the tissue. For the two working electrodes, one was applied with voltage (*V* = *V*(*k*)), and the other one was set to 0. Electrical insulation conditions were applied to the surface in contact with the catheter and the outer surface of the phantom, as it was exposed to pure water with low electrical conductivity (2 × 10^−4^ S/m) [35].

To quickly reach *T_target_*, avoid overshooting, and achieve a small steady-state error, the maximum voltage (30.9 (V)) was used when the temperature difference ($e\left(k\right)$) between the target temperature and the measured temperature was larger than 4 °C. When the difference is less than 4 °C, the voltage *V*(*k*) was calculated with the PID algorithm, which ranged from 13.5 (V) to 30.9 (V). The lower and upper limits of the voltage were manually adjusted. The PID-controlled voltage (*V*(*k*)) was calculated using the following formulas, Equations (2)–(4):(2)$$e\left(k\right)=T_{target}\left(k\right)-T_{0}\left(k\right)\;$$
(3)$$u\left(k\right)=K_{p}e\left(k\right)+K_{i}\overset{k}{\underset{j=0}{\sum }}e\left(j\right)+K_{d}\left(e\left(k\right)-e\left(k-1\right)\right)$$
(4)$$V\left(k\right)=\left\{\begin{array}{c} \;\;\;30.9;\;\;\;\;\;\;\;\;\;\;\;\;\;\;\;\;\;\;\;u\left(k\right)\geq 17.4\\ u\left(k\right)*0.95+13.5\;;\;\;\;\;\;\;\;\;\;0<u\left(k\right)<17.4\\ 13.5;\;\;\;\;\;\;\;\;\;\;\;\;\;\;\;\;\;\;\;u\left(k\right)\leq 0 \end{array}\right.$$
where *T_target_* and $T_{0}\left(k\right)$ are the target temperature and the measured temperature in each sample, respectively. $u\left(k\right)$ (V) is the control function in each sample. The proportional gain ($K_{p}$), integral gain ($K_{i}$), and derivative gain ($K_{d}$) were equal to 1.30 (V/K), 0.19 (V/K), and 0 (V/K), respectively. $K_{p}$*,*
$K_{i}$*,*
$K_{d}$ and the range of *u*(*k*) in Equation (4) were set to be the same as in the experiment, which was manually adjusted to ensure reaching the target temperature quickly (<10 s) and to keep steady with a control precision of ±0.5 °C. In the experiment (Section 2.1), $K_{d}$ was set to 0.56 (V/K) to avoid overshooting, given the response times of the thermocouples.

The volumetric heat generation rate ($Q_{ec}$) (W/m^3^) was calculated using Equation (5):(5)$$Q_{ec}=\sigma E^{2}$$
where E is the root-mean-square value of the electric field intensity (V/m) and σ is the electrical conductivity (S/m).

The heat transfer governing equation is given as follows, Equation (6):(6)$$\rho c\frac{\partial T}{\partial t}=\nabla \cdot \left(k\nabla T\right)+Q_{ec}\;$$
where ρ, c, and k are the density (kg/m^3^), the specific heat capacity (J/(kg·K)), and the thermal conductivity (W/(m·K)), respectively. The electrical and thermal properties of the phantom, the polyimide catheter, and the copper electrodes are listed in Table 1.

The convective thermal boundary was applied to the inner surface of the catheter to simulate the flow of cooling water in the catheter, Equation (7):(7)$$Q_{f}=h\left(T_{w1}-T_{f}\right)\;$$
where *h* (W/(m^2^·K)) is the convective heat transfer coefficient and $T_{w1}$ and $T_{f}$ are the temperature of the inner wall of the balloon catheter and the cooling water temperature. To simulate the in vivo thermal treatment, the initial temperature was set to 37 °C and the outer surface of the phantom was set as thermal insulation. 

### 2.3. Optimal Heating Strategy 

To further determine the heating strategy for the conformal treatment of the fibrotic plaques, an optimization algorithm was proposed based on the quantitative relationships between the ablation dimensions and different combinations of the parameters, which were obtained from the numerical model.

## 3. Results

### 3.1. Influence of Parameters Based on Experiments 

Figure 5a illustrates the temperature at three monitoring points with the target temperature (*T_target_*), the cooling water temperature (*T_f_*), and the flow velocity (*V_f_*) set to be 38 °C, 20 °C, and 2.85 m/s, respectively. It can be seen that the temperature (T0) at the control point reached the setpoint (38 °C) in less than 10 s. As seen in Figure 5a, T1, which represented a point in the plaque, had the highest temperature, while T0, which was located on the inner surface of the phantom, was the lowest. The temperatures of T2 (46.11 ± 0.01 °C) and T3 (40.89 ± 0.02 °C) were much lower than that in the plaque (T1: 62.12 ± 0.05 °C), over 16–22 °C. As temperatures T1–T3 reached equilibrium in less than 45 s, the final equilibrium temperatures under different conditions were used and are shown in Figure 5b–d. 

Figure 5b illustrates the steady-state temperatures at the measured points (T1–T3) with different target temperatures (*T_target_*). It was found that, for all three points, the final temperature increased linearly with *T_target_* (R^2^ > 99.5%). The steady-state temperatures at these points were found to decrease linearly with the cooling water temperature (*T_f_*) (Figure 5c) and increase slightly with the flow rate (*V_f_*) (Figure 5d). 

The temperature of TC1, which represented the plaque area, changed with *T_target_*, *T_f_*, and *V_f_*. In particular, the range of the T1 temperature could be adjusted by more than 25 °C by adjusting *T_target_* or *T_f_* (Figure 5b,c), while it could be adjusted by less than 5 °C by adjusting *V_f_* (Figure 5d). *T_target_* and *T_f_* were the two major factors affecting the steady temperature in the phantom (T1–T3, especially T1).

However, only the temperatures at certain points were obtained through the experimental measurement. To determine the temperature distribution, especially to define the ablation range, numerical results are obtained.

### 3.2. Numerical Modeling

The numerical model was validated using the tissue-mimicking phantom heating experiments. The comparison of the numerical calculation results with the experimental results is presented in Figure A1, Figure A2 and Figure A3 (*T_target_* = 40 °C, *T_f_* = 15 °C, and *V_f_* = 2 m/s). A one-way sensitivity analysis was performed to verify the robustness of the model [39]. It was found that when the electrical conductivity increased by 20% from 0.30 S/m to 0.36 S/m, the maximum value of temperature T1 (the point located in the plaque) changed from 59.92 °C to 59.93 °C, while when the thermal conductivity increased by 20% from 0.51 W/(m∙K) to 0.63 W/(m∙K), the maximum T1 changed from 58.96 °C to 61.13 °C, which was less than 3%. The convergence of the model was verified, as shown in Figure A5.

The relationships between the lesion dimensions and the two main control parameters were investigated. Moreover, to evaluate the evolution of absorbed RF energy during the two-minute heating process, the SAR (specific absorption rate) distribution was also calculated.

#### Numerical Results

A 50 °C isothermal curve was used to determine the ablation region, as 50 °C has been found to be the lethal temperature for arterial-related cells [24]. The ablation depth and protection depth were defined as the maximum and minimum distances of a 50 °C isothermal away from the outer surface of the catheter.

(a) The evolution of temperature/SAR distribution

The evolution of the ablation zone during a two-minute heating process is shown in Figure 6a–d, with *T_target_* = 35 °C, *T_f_* = 18.5 °C, and *V_f_* = 2 m/s (*h* = 9303 W/(m^2^*K)). The ablation zone (marked with a green line) appeared approximately 0.25 mm away from the inner surface of the phantom after 12 seconds of heating, and it was initiated from the two corners between the paired electrodes with the shapes of raindrops (Figure 6a). Then, the two raindrops fused quickly and extended further (Figure 6b,c). The ablation depth increased with the heating time, while the protection depth decreased with time. After 30 s, the 50 °C isothermal regions, including the ablation depth and the protection depth, reached a steady state (Figure 6d). 

To analyze the deposition of RF energy in tissue during the heating process, the SAR distribution is shown in Figure 6e–h. The distribution of SAR was found to mainly concentrated in the region close to the working electrodes, and it hardly changed with time though the electrical conductivity changes with the temperature. A comparison of Figure 6a–d with Figure 6e–f shows the influence of the surface convection effect. Volumetric heating (SAR) was established quickly in the tissue. It was the propagation of surface cooling that changed the temperature profile in the tissue.

(b) The effects of parameters on the dimensions of the lesions

The target temperature (*T_target_*) and the cooling agent temperature (*T_f_*) were the two critical parameters regulating the temperature profile based on the previous experiment (Figure 5). The local temperature in the tissue increased with *T_target_
*and decreased with the flow temperature (*T_f_*). To determine how the ablation zone varied with these two control parameters, different values of *T_target_
*and *T_f_
*were simulated, and the results are shown in Figure 7. With an increase in *T_target_*, the ablation zone expanded from the intermediate region, while with an increase in *T_f_* the ablation zone shrank. The two parameters had contrary effects on the ablation depth (*A*) and the protection depth (*P*). With a single control parameter, the ablation depth (*A*) and protection depth (*P*) changed synchronously, and it was impossible to realize the conformal control target for the diseased blood vessel. The two control parameters may need to be used simultaneously.

Therefore, to further quantitatively establish the relationship between the ablation zone geometry and the two parameters, a parameter sweep analysis was performed. The target temperature (*T_target_*) ranged from 29 °C to 43 °C, and the coolant temperature circulating inside the RF balloon catheter (*T_f_*) ranged from 0.5 °C to 36.5 °C at an interval of 1 °C. *V_f_* was kept at 2 m/s in all conditions. The ranges of the parameters were chosen to protect the endothelium from excessive heating.

Among the sweeping cases, if the temperature of the endothelial layer was below 50 °C and there were regions above 50 °C, it was defined as a feasible case, and the combined condition was labeled in green, as shown in Figure 8. The cases with excessive heating, where temperatures close to the endothelial layer were all above 50 °C, were labeled in red. The cases with no temperature higher than 50 °C were labeled in blue. The green diagram then clearly labeled all possible combined conditions that could be used to realize the control of both the protection depth and the ablation depth. It could also be seen that the green diagonal area with feasible cases, which accounted for 16.2% (90/555) of the total cases, was relatively narrow.

To determine the control ranges of these two parameters, the relationships between the two parameters and the ablation depth (*A*) as well as the protection depth (*P*) in all feasible cases were analyzed, as shown in Figure 9. It can be seen that the ablation depth (*A*) ranged from 0.47 mm to 1.43 mm, while the protection depth (*P*) ranged from 0 mm to 0.26 mm, which covered the range for most arterial plaques [40,41], such as in peripheral [22], brachial, and radial arteries [23]. 

### 3.3. Algorithm of Optimal Heating Strategy

With the obtained quantitative relationships between the ablation depth (*A*), the protection depth (*P),* and the two parameters (*T_target_* and *T_f_*), an optimal heating strategy was then established to investigate the possibility of the bidirectional control of ablation. Two implicit discrete functions describing the relationship, *A* = ***M*** (*T_f_*, *T_target_*) and *P* = ***N*** (*T_f_*, *T_target_*), were obtained from the above parameter sweep analysis. For the excessive ablation cases and insufficient ablation cases, the values of the protection depth or ablation depth were set to zero in ***M*** (*T_f_, T_target_*) and ***N*** (*T_f_*, *T_target_*), respectively. Then, the control problem was simplified to an optimization problem with boundaries. The bilinear interpolation and iteration from these two functions were proposed based on the linear relationship between ablation depth/protection depth and *T_target_*/*T_f_* in the region. The targeted ablation depth (*A_t_*) and the protection thickness (*P_t_*) were used as input variables. The two existing discrete functions (***M*** and ***N***) were used to calculate the objective function $\left\{\left\|A_{t}-M\|+\left(A_{t}/P_{t}\right)\|P_{t}-N\right\|\right\}$. Before starting iterative interpolation, the initial values *T_f,_*_0_ and *T_target,_*_0_ were obtained by calculating the function min$\left\{\left\|A_{t}-M\|+\left(A_{t}/P_{t}\right)\|P_{t}-N\right\|\right\}$. Then, *A_0_* and *P_0_* were obtained from the functions ***M*** (*T_f,_*_0_, *T_target,_*_0_) and ***N*** (*T_f,_*_0_, *T_target,_*_0_). *A*_0_ and *P*_0_ were compared with the targeted values of *A_t_
*and *P_t_*. If the relative errors were less than 5% for both the protection depth and the ablation depth, the iteration continued. Two terminal conditions were used to stop the iteration: (1) if the objective function $\left\{\left\|A_{t}-A_{k}\|+\left(A_{t}/P_{t}\right)\|P_{t}-P_{k}\right\|\right\}$ was less than 0.01 mm and (2) if the iteration number (k) reached 10. During the iteration, *T_target,k_* and (or) *T_f,k_* were adjusted according to the distances between the current values (the ablation depth (*A_k_*) and the protection depth (*P_k_*)) and the targeted values (*A_t_* and *P_t_*) to obtain new *T_target, k+_*_1_ and *T_f, k+_*_1_ values. ***M*** and ***N*** were linearly interpolated bidirectionally from ***M*** (*T_target,k,_ T_f,k_*) and ***N*** (*T_target,k,_ T_f,k_*) to calculate the new corresponding ablation depth *(A_k+_*_1_) and protection depth (*P_k+_*_1_). Then, *k*, *A_k_*, and *P_k_,* as well as ***M*** and ***N***, were updated, and the above procedures were repeated. A detailed algorithm flowchart is shown in Figure 10.

Two cases with different treatment requirements were used to illustrate the above algorithm. For case 1, the objective ablation depth and protection depth were set to be 0.78 mm and 0.06 mm (Figure 11a). According to the algorithm, the control parameters were then determined to be *T_target_* = 41.9 °C and *T_f_* = 33.6 °C. With these two parameters, the ablation geometry was calculated using the model described above. The ablation depth was found to be 0.78 mm, and the protection was found to be 0.06 mm. As the intimal thickness increased (1.02 mm in ablation depth and 0.1 mm in protection depth), as shown in case 2 (Figure 11b), *T_target_* = 29.9 °C and *T_f_* = 8.6 °C were obtained with the algorithm, and there were errors of 0.01 mm in both ablation depth and protection depth. 

## 4. Discussion

Based on the experimental study, it was found that the temperatures at the three monitoring points (T1, T2, and T3) inside the tissue, representing the locations of plaque, adventitia, and surrounding tissue [28], changed linearly with the three control parameters. Moreover, the two parameters *T_target_* and *T_f_* were found to be the main factors that adjusted the ablation results, including the ablation depth and protection depth. However, the RF volumetric heating power and the convection cooling power were simultaneously affected by *T_target_* and *T_f_*. The ablation depth and protection depth changed synchronously with a single control parameter. Only through the simultaneous control of the two parameters could bidirectional control of confined penetrating heating be realized. 

Current RF electrodes and the cooling agent setup were found to be able to cover an ablation depth ranging from 0.47 mm to 1.43 mm, with a protection depth ranging from 0 mm to 0.26 mm, which is sufficient for most arterial plaques [40,41]. Further explorations into increasing this range would help broaden its application in clinical settings. 

The proposed algorithm combining bilinear interpolation and iteration was used to obtain the optimal combination of *T_target_* and *T_f_* for the targeted ablation depth and protection depth, which is essentially an inverse problem of heat transfer [42,43]. In our study, this was simplified as a multivariable optimization problem, and the results of the numerical calculations were used to simplify the process of determining physical relations. The algorithm illustrated the feasibility of using the two proposed parameters for the bidirectional control of the arterial plaque treatment. Though there were errors of 0.01 mm in control for both the ablation depth and the protection depth, as long as the resulting protection region is larger than the objective, with a sufficient ablation depth, this should be clinically acceptable.

### Limitation 

There are certain limitations to this study. Though a one-way sensitivity analysis verified the robustness of the current model, a model that takes into account the differences in the electrical and thermal properties of heterogeneous plaques would give more precise predictions, especially for advanced atherosclerotic plaques, which include lipids and calcified regions [44,45]. The feasibility of the bidirectional control strategy using two main control parameters, *T_target_* and *T_f_*, was proven with numerical calculations. Further precision imaging of the ablation, together with careful experimental design, is needed.

## 5. Conclusions

The temperature of the control point on the inner surface of the endothelial layer (*T_target_*) and the temperature of the cooling flow (*T_f_*) were found to be two effective factors for adjusting the ablation results in the newly proposed radiofrequency balloon angioplasty. The two parameters have contrary effects on the control of the ablation depth and protection depth when used alone. An ablation range covering most clinical plaque treatment requirements can be attained through the simultaneous use of the two parameters. The feasible combinations of the conditions were found. A bidirectional penetrating heating strategy can be realized with the proposed bilinear interpolating and iterating algorithm. Future studies to widen the control range while considering the complexity of real plaque properties would help with the technique’s further application.

## Figures and Tables

**Figure 1 bioengineering-10-00162-f001:**
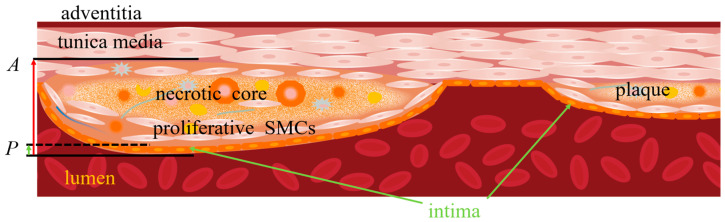
Illustration of two atherosclerotic lesions with different geometries. *A* and *P* define the ablation depth and the protection depth.

**Figure 2 bioengineering-10-00162-f002:**
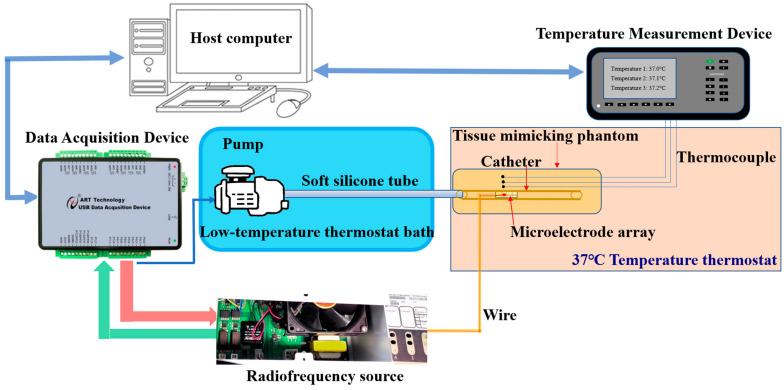
Illustration of the tissue-mimicking phantom heating experiment. The radiofrequency catheter system consisted of an RF source and a microelectrode array, The cooling unit (an adjustable pump and a low-temperature thermostat bath), the control unit (DAQ and host computer), and the tissue-mimicking phantom were immersed in a 37 °C thermostat bath. A temperature measurement device connected to the host computer was used to measure, display, and record the temperature.

**Figure 3 bioengineering-10-00162-f003:**
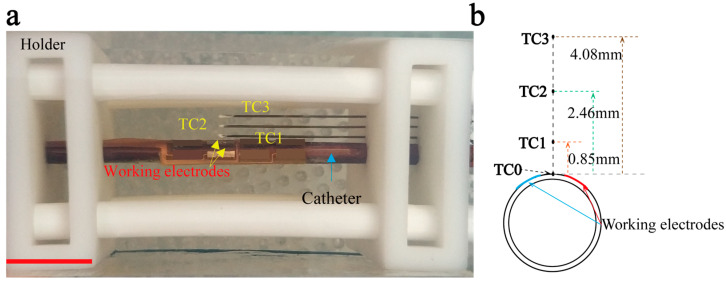
(**a**) Picture of the phantom with thermocouples (TC1, TC2, and TC3) and catheter. Scale bar: 1 cm. (**b**) The relative positions among the thermocouples, working electrodes, and catheter. Thermocouple TC0 measured the control point temperature. TC0, TC1, TC2, and TC3 represented the locations of the endothelial lining, plaque, adventitia, and surrounding tissue.

**Figure 4 bioengineering-10-00162-f004:**
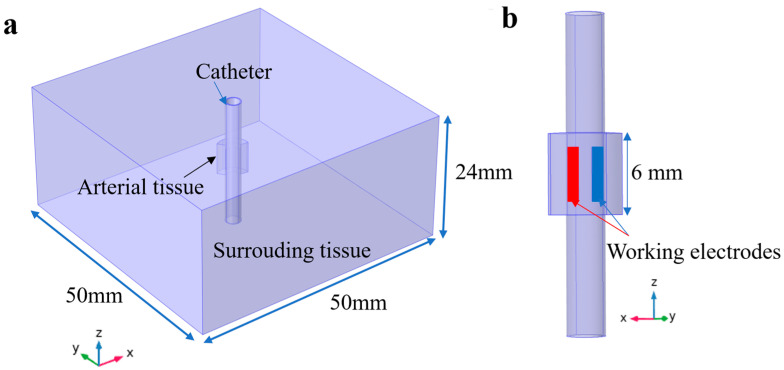
(**a**) The geometry setup of the model. The large block is the surrounding tissue. The small portion is the arterial tissue. The cylinder is the catheter. (**b**) An enlarged illustration of the catheter and the arterial tissue near the paired electrodes.

**Figure 5 bioengineering-10-00162-f005:**
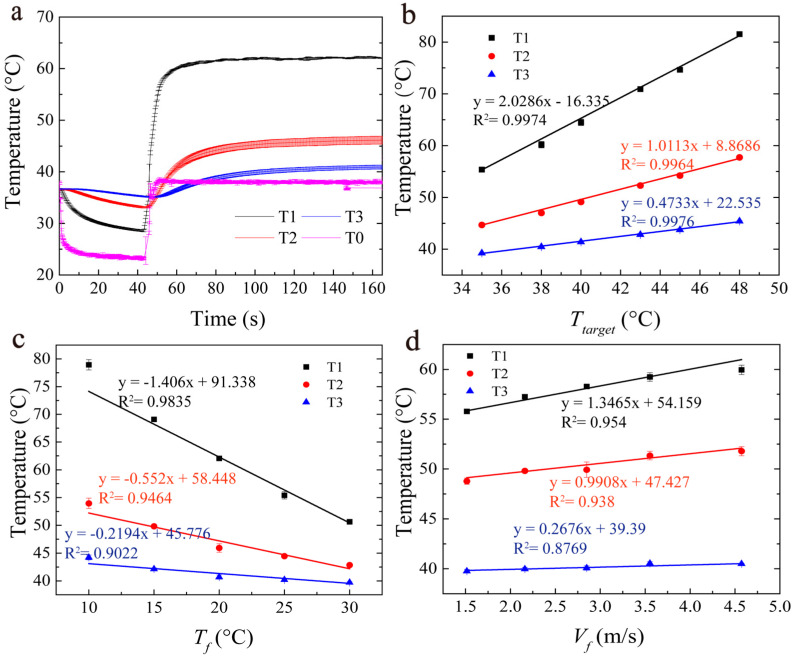
(**a**) The temperature evolution of T0 and T1–T3 in the phantom (*T_target_* = 38 °C, *T_f_* = 20 °C, *V_f_* = 2.85 m/s). (**b**–**d**) T1–T3 in the equilibrium stage vs. the three control parameters: *T_target_*, *T_f_*, and *V_f_*. (**b**) The target temperature (*T_f_* = 15 °C, *V_f_* = 2.85 m/s), (**c**) the cooling water temperature (*T_target_* = 38 °C, *V_f_* = 2.85 m/s), and (**d**) the cooling water velocity (*T_target_* = 38 °C, *T_f_* = 20 °C). T1, T2, and T3: the temperatures measured by TC1, TC2, and TC3. T0: the temperature at the control point. All of the data were collected in triplicate and expressed as the means ± SDs.

**Figure 6 bioengineering-10-00162-f006:**
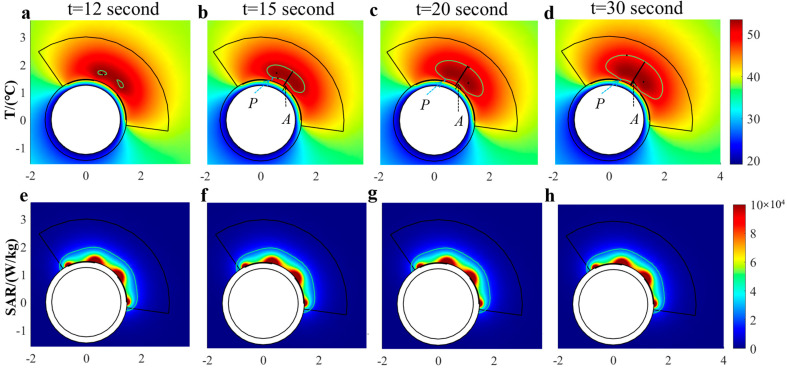
(**a**–**d**) Evolution of the ablation zone and (**e**–**h**) SAR distribution during the heating process (*T_target_* = 35 °C, *T_f_* = 18.5 °C, *V_f_* = 2 m/s, and *h* = 9303 W/(m^2^*K)). Ablation depth (*A*) and protection depth (*P*) are marked in (**b**).

**Figure 7 bioengineering-10-00162-f007:**
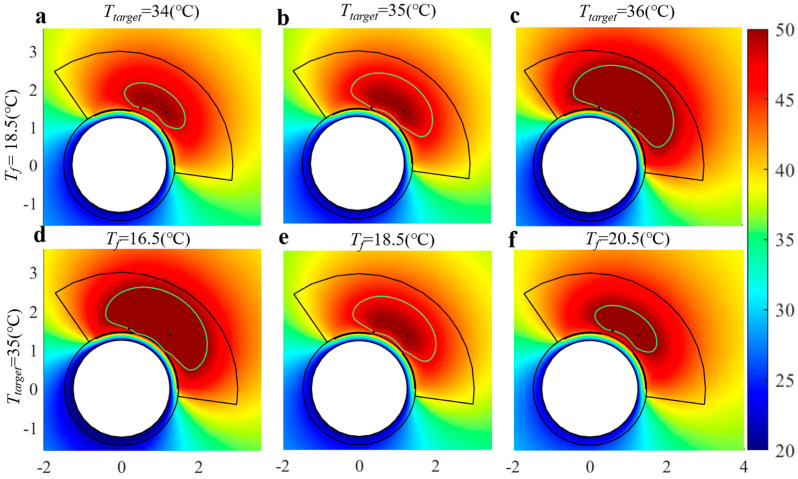
The ablation zone (marked with a green line) varied as *T_target_* (**a**–**c**) or *T_f_* (**d**–**f**) increased. *V_f_* = 2 m/s in all conditions.

**Figure 8 bioengineering-10-00162-f008:**
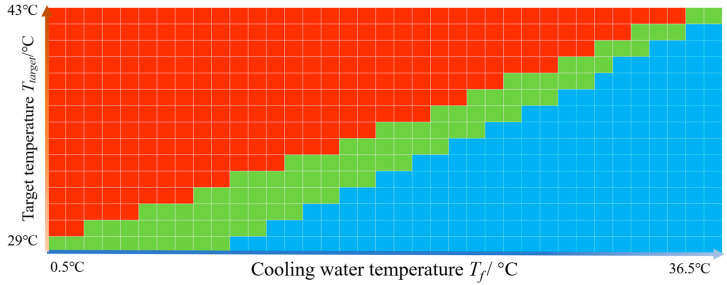
The combined conditions that were feasible for the control of conformal heating. Feasible cases are shown in green, in which the temperature close to the inner surface was below 50 °C, while the temperatures in the deeper sites were above 50 °C. The excessive heating cases are shown in red, where the temperature close to the tissue’s inner surface was above 50 °C. The insufficient cases are shown in blue, where the temperatures in all regions were below 50 °C. The target temperature (*T_target_*) ranged from 29 °C to 43 °C, and the cooling water temperature (*T_f_*) ranged from 0.5 °C to 36.5 °C.

**Figure 9 bioengineering-10-00162-f009:**
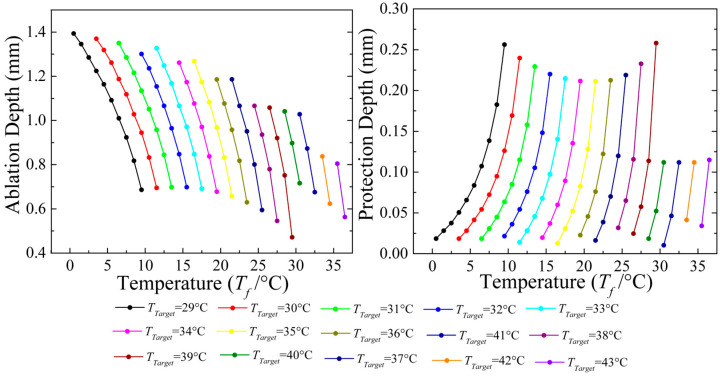
The relationships of the two parameters with the ablation depth (**left**) and the protection depth (**right**) in the feasible cases.

**Figure 10 bioengineering-10-00162-f010:**
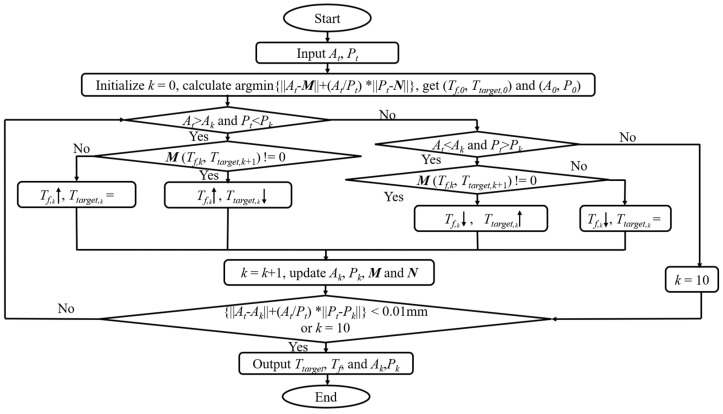
Algorithm flowchart determining *T_target_* and *T_f_* for desired ablation depth (*A_t_*) and protection depth (*P_t_*). ***M*** and ***N*** are the functions of ablation depth and protection depth change with *T_target_* and *T_f_*, respectively. *k* is the number of iterations. *A_k_*, *P_k_*, *T_f,k_*, and *T_target,k_
*are the *k_th_* calculations of *A*, *P*, *T_f_,* and *T_target_*.

**Figure 11 bioengineering-10-00162-f011:**
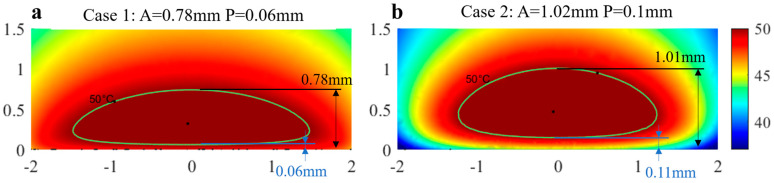
(**a**) Case 1: the objective ablation parameters were *A* = 0.78 mm and *P* = 0.06 mm; (**b**) Case 2: *A* = 1.02 mm and *P* = 0.1 mm. The real ablation parameters are marked with arrows.

**Table 1 bioengineering-10-00162-t001:** The electrical and thermal parameters used in the model.

Parameter	Tissue	Polyimide	Copper
Thermal conductivity, *k* (W/(m·K))	0.60 + 0.0014(*T* − *T_ref_*) [13]	0.25 [36]	386.47
Density, ρ (kg/m^3^)	998.2	2200	8935.4
Specific heat capacity, c (J/(kg·K))	4183	1050	383.9
Electrical conductivity, σ (S/m)	0.33 + 0.0035(*T* − *T_re__f_*)) [37,38]	0	5.76 × 10^7^

*T_ref_* = 20 °C.

## Data Availability

The datasets created and/or analyzed during the current investigation are available upon reasonable request from the corresponding author. All figures in this paper are original.

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
