# Peer review of "A New Conformal Penetrating Heating Strategy for Atherosclerotic Plaque"

_bioengineering, 2023, doi:10.3390/bioengineering10020162_

Round 1
Reviewer 1 Report
Nice review with appealing images and text. Please revise text for language
Author Response
Point 1: Nice review with appealing images and text. Please revise text for language
Response 1: Thanks for the reviewer’s suggestion. The manuscript has been carefully re-edited. Please see the revised manuscript.
Reviewer 2 Report
1、 Line 92, there are four electrodes in total. Should the number of electrodes used to be determined according to the size and location of the plaque? What is the purpose of using only one pair? How to determine the position of this pair of electrodes? In Fig. 3 (b), the included angle between the two electrodes is about 60 °, but if the position relationship of the electrodes and the number of electrodes change, will the overall conclusion be affected?
2、 Line 185-188, why the temperature difference of 4 ℃ is selected as the regulating parameter.
3、 Line 192, what are the selection criteria for these three parameters? Is this group of parameters the best choice to ensure that the target temperature can be reached quickly, kept stable, and the control accuracy is ± 0.5 ℃? How to ensure the best choice?
4、 Line 349, what is the specific difference between the two methods of stopping iteration under terminal conditions? Can different parameter adjustment processes ensure that the final result is the same parameter?
5、 If the shape of the plaque is irregular and the depth is inconsistent, how can this problem be solved by the method in this paper?
Author Response
Point 1: Line 92, there are four electrodes in total. Should the number of electrodes used to be determined according to the size and location of the plaque? What is the purpose of using only one pair? How to determine the position of this pair of electrodes? In Fig. 3 (b), the included angle between the two electrodes is about 60 °, but if the position relationship of the electrodes and the number of electrodes change, will the overall conclusion be affected?
Response 1: Yes, 4 electrodes surrounding the balloon are used according to the size and location of the plaque. When in the real application, different numbers of electrodes may be manufactured on the surface of the balloon, and the selection of the balloon with a certain number of electrodes should be determined according to the size and location of the plaque. In this study, to achieve the bidirectional depth control of the penetrating thermal treatment of the plaques, we used bipolar electrodes with a focus on the treatment depths. The treatment range in the peripheral and the axial direction could be adjusted by selecting different numbers of electrodes and individual control on each electrode, which will be studied in the future.
The position of this pair of electrodes can be determined by the guidance of intravascular imaging, i.e OCT, IVUS.
The conclusions of this study include the influence of these control parameters (Ttarget, Tf, Vf), the ranges of the ablation depth and the protection depth, and the proposed algorithm to provide the optimal heating strategy. If the position relationship of the electrodes and the number of electrodes change, the influence of these control parameters should be the same, such as regulating the ablation depth and protection depth by adjusting Ttarget and Tf, which is inferred according to the previous experience. The numerical model can still be used for the new electrode parameters and the proposed algorithm can be used to provide the optimal strategy, though the ranges of the ablation depth and the protection depth are different.
Point 2: Line 185-188, why the temperature difference of 4 ℃ is selected as the regulating parameter.
Response 2: The temperature difference of 4 °C is selected empirically to ensure a fast and steady control of the heating process. As a higher temperature difference brings a larger steady-state error while a lower temperature difference increases the risk of the overshoot, it is decided manually. Lines 191 to 192.
Point 3: Line 192, what are the selection criteria for these three parameters? Is this group of parameters the best choice to ensure that the target temperature can be reached quickly, kept stable, and the control accuracy is ± 0.5 ℃? How to ensure the best choice?
Response 3: The selection criteria of these three parameters is to make the temperature of the control point reach the target temperature as quickly as possible and keep it stable. Yes, through the parameter sweeping in the experiment, we found that this group of parameters is the best choice that can reach the target temperature within 10s, with a small overshoot and a small stability error.
Point 4: Line 349, what is the specific difference between the two methods of stopping iteration under terminal conditions? Can different parameter adjustment processes ensure that the final result is the same parameter?
Response 4: The first method of stopping iteration is to compare the objective function error of each iteration with the minimum error, 0.01mm. The second method of stopping iteration is to judge whether the current iteration number reaches the maximum number of iterations,10. The specific difference between the two methods of stopping iteration under terminal conditions is the number of iterations and the corresponding computation time. If the objective function error is less than 0.01mm, the iterative computation ends early.
Yes, we believe that parameters found with different methods should be similar though there may be some error depending on the method itself as the proposed algorithm is a method to quickly find the most suitable parameters which already exist in the solutions instead of figuring out the functions themselves.
Point 5: If the shape of the plaque is irregular and the depth is inconsistent, how can this problem be solved by the method in this paper?
Response 5: For the irregular plaque with inconsistent depth, multiple micro-electrodes shall be used, each bipolar ablation from the paired microelectrodes as found by the strategy developed in this study can be used as one unit (one scanning spot), and through controlling the on-off state of the multiple microelectrode pairs by the guidance of imaging modality, ab irregular ablation shape may be resulted with the plaque being fully covered.

Reviewer 3 Report
In this contribution, experimental and numerical studies have been presented for tissue-mimicking phantom heated devices, that are representative of an atherosclerotic plaque under penetrating heating. After describing the experimental setup to run experiments, numerical model development is shown by making references to a 3D geometry where Laplace and energy equation are used to compute voltage and temperature evolution, respectively. Results are shown under different scenarios by varying inlet flow velocity, temperature, as well as other control parameters. Predictions have been shown to be in good agreement with experiments; furthermore, the proposed control algorithm showed the input parameters that allow to cover the desired ablation range depending on the clinical treatment to be done.
The reviewer thinks that this is an exhaustive contribution with both experimental and predictive approaches, that covers a new potential solution to improve atherosclerosis plaque treatment based on induced heating. It is suggested to consider the present paper after addressing the following points
- In order to stress the novelty from the present paper, in the introduction please underline differences between the present contribution and [1]
- When the authors describe how did they generate the tissue-mimicking phantom in terms of chemical composition etc., did they check that electrical properties of the phantom are consistent with real tissues? This aspect might be relevant in similar studies [2]
- Authors employed velocities that vary from 1.51 up to 4.57 m/s. How did the authors decide these values?
- In terms of modeling strategy (subsection 2.2), which are the differences between this contribution and similar works about bioheat transfer [3-6]? Please report and discuss all this at the beginning of the subsection in order to justify differences between the present contribution and such similar works [3-6]
- Why the volumetric heat generation in Eq. 5 term is not divided over a factor of 2 as often happens in similar study [7]?
- It seems that density is missing in Eq. 6. Please correct this
- In Eq. 7, the heat transfer coefficient employed are reported in the appendix A. Did the authors check that the correlations employed (appendix A) are appropriate in terms of flow regime (laminar/turbulent), regions (entry, thermally developed etc), and so on? Furthermore, is it appropriate to assume thermal insulation for the phantom outer surfaces? Aren't any relevant heat losses to be accounted here?
- Temperatures reached in this study last in between 20 and 80 °C approximately. Did the authors consider density and specific heat capacity variation with temperature too (Table 1)?
- Comparisons between experimental and numerical data are reported in Figure A1. There is a good agreement between experiments and predictions, but why it seems that predictions are often overpredicting temperature evolution? Is there something like heat losses to be accounted for that makes temperatures smaller in experiments?
- In Figure A4, aren't the elements too few? It seems that a very low number of elements has been used here
- Please adjust references by making references to the authors' guidelines
[1] Zhao, S., Zou, J., Wang, H., Qin, J., Lu, X., Zhang, A., & Xu, L. X. (2020). A new radiofrequency balloon angioplasty device for atherosclerosis treatment. BioMedical Engineering OnLine, 19(1), 1-18.
[2] Ortega-Palacios, R., Trujillo-Romero, C. J., Cepeda Rubio, M. F. J., Vera, A., Leija, L., Reyes, J. L., ... & Vega-López, M. A. (2018). Feasibility of using a novel 2.45 GHz double short distance slot coaxial antenna for minimally invasive cancer breast microwave ablation therapy: Computational model, phantom, and in vivo swine experimentation. Journal of Healthcare Engineering, 2018.
[3] Iasiello, M., Vafai, K., Andreozzi, A., & Bianco, N. (2019). Hypo-and hyperthermia effects on LDL deposition in a curved artery. Computational Thermal Sciences: An International Journal, 11(1-2).
[4] Iasiello, M., Andreozzi, A., Bianco, N., & Vafai, K. (2019). The porous media theory applied to radiofrequency catheter ablation. International Journal of Numerical Methods for Heat & Fluid Flow.
[5] Ortega-Palacios, R., Trujillo-Romero, C. J., Cepeda-Rubio, M. F. J., Leija, L., & Vera Hernández, A. (2020). Heat transfer study in breast tumor phantom during microwave ablation: Modeling and experimental results for three different antennas. Electronics, 9(3), 535.
[6] Trujillo, M., Prakash, P., Faridi, P., Radosevic, A., Curto, S., Burdio, F., & Berjano, E. (2020). How large is the periablational zone after radiofrequency and microwave ablation? Computer-based comparative study of two currently used clinical devices. International Journal of Hyperthermia, 37(1), 1131-1138.
[7] Keangin, P., & Rattanadecho, P. (2018). A numerical investigation of microwave ablation on porous liver tissue. Advances in Mechanical Engineering, 10(8), 1687814017734133.
Round 2
Reviewer 3 Report
In the reviewer's opinion, the paper can be accepted